# Lateral advection supports nitrogen export in the oligotrophic open-ocean Gulf of Mexico

Thomas B. Kelly [1,2,5✉], Angela N. Knapp [1], Michael R. Landry[3], Karen E. Selph[4], Taylor A. Shropshire[1,2], Rachel K. Thomas [1] & Michael R. Stukel[1,2]

In contrast to its productive coastal margins, the open-ocean Gulf of Mexico (GoM) is notable for highly stratified surface waters with extremely low nutrient and chlorophyll concentrations. Field campaigns in 2017 and 2018 identified low rates of turbulent mixing, which combined with oligotrophic nutrient conditions, give very low estimates for diffusive flux of nitrate into the euphotic zone (< 1 μmol N m$^{-2}$ d$^{-1}$). Estimates of local N$_2$-fixation are similarly low. In comparison, measured export rates of sinking particulate organic nitrogen (PON) from the euphotic zone are 2 – 3 orders of magnitude higher (i.e. 462 – 1144 μmol N m$^{-2}$ d$^{-1}$). We reconcile these disparate findings with regional scale dynamics inferred independently from remote-sensing products and a regional biogeochemical model and find that laterally-sourced organic matter is sufficient to support >90% of open-ocean nitrogen export in the GoM. Results show that lateral transport needs to be closely considered in studies of biogeochemical balances, particularly for basins enclosed by productive coasts.

[1] Department of Earth, Ocean and Atmospheric Sciences, Florida State University, Tallahassee, FL, USA. [2] Center for Ocean-Atmospheric Prediction Studies, Florida State University, Tallahassee, FL, USA. [3] Integrative Oceanography Division, Scripps Institution of Oceanography, La Jolla, CA, USA. [4] Department of Oceanography, University of Hawaii at Manoa, Honolulu, HI, USA. [5] Present address: College of Fisheries and Ocean Science, University of Alaska Fairbanks, Fairbanks, AK, USA. ✉email: tbkelly@alaska.edu

Conventionally, primary production is partitioned between new and regenerated production based on the source of inorganic nitrogen:[1] new production (NP) is fueled by N (nitrogen) input from external sources (e.g., upwelled nitrate and $N_2$-fixation) while regenerated production (RP) comes from the internal recycling of nitrogen (e.g., ammonium). While RP often supports the vast majority of total primary production in open-ocean ecosystems, new nitrogen sources (NP) are necessary to support nitrogen and carbon export into the deep sea because the mass fluxes into and out of a system must balance when integrated over sufficiently large spatiotemporal scales. While the NP and RP dichotomy is well established in the simple biogeochemical metric of the $f$-ratio ($=NP/(NP+RP)$), direct measurement of NP is complicated by the intrinsic complexities of nitrogen cycling. For example, the most common NP measurement technique, nitrate uptake incubations, assumes negligible rates of nitrification (i.e., oxidation of ammonium into nitrate) within the euphotic zone. Nitrification, which regenerates nitrate from ammonium, may at times be a dominant source of nitrate in surface waters[2–5], especially in oligotrophic waters where ammonium concentrations can exceed those of nitrate. Additionally, strong gradients in organic matter concentrations between coastal and offshore environments provide significant lateral transport potential. Through lateral transport, nutrients and organic matter can be carried into the offshore region, although this flux is seldom included in NP calculations. Modeling studies have suggested that net offshore fluxes can be an important addition to regional N balances, supporting 24–80% of total export in subtropical gyres[6,7]. However, lateral transport is difficult to constrain because spatiotemporal scales of advection are usually poorly matched to those of direct field measurements.

While numerous biogeochemical budgets have been reported for the shallow shelves of the GoM[8–11], few studies have focused on the oceanic region that covers ~2/3rds of the GoM surface area. Data presented here were collected during two field campaigns of Bluefin Larvae in Oligotrophic Ocean Foodwebs: Investigating

Nutrients to Zooplankton in the GoM (BLOOFINZ-GoM) program targeting the central GoM spawning grounds of the endangered Atlantic Bluefin tuna[12,13]. Major euphotic zone C and N cycling processes were quantified during five Lagrangian studies[14–17], allowing us to compile comprehensive biogeochemical budgets. Here, we combine these in situ measurements with independent remote-sensing observations and a biogeochemical model to test alternate hypotheses that (1) locally upwelled nitrate, (2) $N_2$-fixation, or (3) lateral transport support N export from oligotrophic oceanic waters of the GoM.

## Results and discussion

**Field observations and dynamics**. Nitrate+nitrite (hereafter nitrate) and ammonium were generally < 100 nM throughout the euphotic zone, with nitrate concentrations between the detection limit (~10 nM) and 50 nM in the upper euphotic zone (UEZ; 0–60 m) (Fig. 1b). Nitrate concentrations remained depressed well below the lower euphotic zone (LEZ), with 2.5–12 μM concentrations at 150 m, consistent with prior characterizations of these water masses (ref. [18] and references therein). Ammonium averaged 60 nM, with no depth trend in the euphotic zone (Supplemental Fig. 1a). UEZ chlorophyll-$a$ was consistently below 0.2 mg m$^{-3}$ (Fig. 1c), and UEZ integrated nitrogen pools did not vary significantly throughout the Lagrangian experiments (Supplemental Fig. 1c–e). Most (62–79%) net primary production (NPP) occurred in the UEZ, with a surface maximum of 270–550 μmol C m$^{-3}$ d$^{-1}$. In the LEZ, including the deep chlorophyll maximum (DCM), NPP declined to ~100 μmol C m$^{-3}$ d$^{-1}$ (Fig. 1d). Nitrate uptake rates were not strongly depth-dependent, with maximum rates typically observed between 20 and 60 m and vertically integrated rates of 440–1400 μmol N m$^{-2}$ d$^{-1}$ in UEZ and 260–2050 μmol N m$^{-2}$ d$^{-1}$ in LEZ. Vertically integrated $f$-ratios were 0.04–0.14 (mean: 0.07) in the UEZ and 0.03–0.44 (mean: 0.14) for the LEZ. Although nitrification was not directly measured, ammonium-specific rates from other oligotrophic

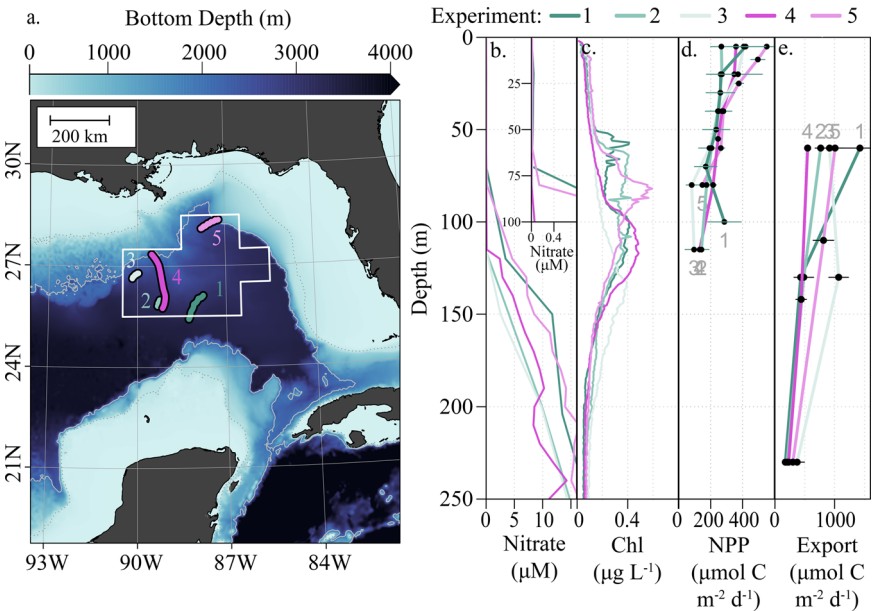

**Fig. 1 Overview of in situ data. a** Study locations in the central Gulf of Mexico (GOM; colored drift trajectories) with a white outline indicating the integration volume from surface to 55 m considered as representative of the eastern Central GoM (depths >2000 m, outside the general flow of the loop current). **b** Vertical profiles of nitrate concentrations with inset detailing low surface concentrations. **c** Vertical profiles of chlorophyll concentrations as recorded by calibrated fluorometer. **d** Vertical profiles of net primary productivity and **e** vertical profiles of particulate nitrogen export at three depths: upper euphotic zone, lower euphotic zone, and 200 m. Colors correspond to Lagrangian experiments, and error bars indicate ±1 SD of means for measurements during experiments.

studies (0.02–0.5 d$^{-1}$; ref. [3] and references within) suggest potential nitrate sources of 1 and 30 μmol N m$^{-3}$ d$^{-1}$, which could supply the majority of nitrate utilized by the phytoplankton community.

Gravitational export of organic matter from the UEZ[14] (at 60 m) ranged from 590 to 1530 μmol N m$^{-2}$ d$^{-1}$ and generally decreased with depth (460–1140 μmol N m$^{-2}$ d$^{-1}$ at the base of the euphotic zone; 190–400 μmol N m$^{-2}$ d$^{-1}$ at 200 m; Figs. 1e and 2a). UEZ export was 164% (range: 90–330%) that of LEZ, indicating significant consumption of sinking particles within the LEZ. Active export of euphotic zone N due to ammonium excretion of migratory zooplankton at depth ranged from 20 to 172 μmol N m$^{-2}$ d$^{-1}$ (mean: 71 μmol N m$^{-2}$ d$^{-1}$; Fig. 2b). This underestimates in situ active transport since organic excretion and zooplankton mortality at depth are not considered[19].

The abundance of *Trichodesmium*[20], a dominant diazotroph in the GoM, ranged from 0 to 19 trichomes L$^{-1}$, which based on a photo-fixation model[21], implies maximum N$_2$-fixation rates of between 0 and 0.38 μmol N m$^{-3}$ d$^{-1}$ and vertically integrated fixation rates of 0.4–2.8 μmol N m$^{-2}$ d$^{-1}$ in the UEZ (Fig. 2d). While lower than in some other regions[22,23], such rates are consistent with Northern GoM findings[15,24] and the low cell densities observed. Furthermore, the isotopic signature of sinking particulate N (i.e., δ$^{15}$N) was nearly identical to that of subsurface nitrate[15], leaving little room in the δ$^{15}$N budget for N$_2$-fixation (see Supplemental Discussion).

Thorpe-scale analysis[25] of vertical eddy diffusivity in the LEZ ranged from 10$^{-6}$ to 10$^{-4}$ m$^2$ s$^{-1}$, which, combined with the small observed gradients in NO$_3^-$ concentration, yields LEZ flux estimates of 0.01–1 μmol N m$^{-2}$ d$^{-1}$ (Fig. 2c; Supplemental Fig. 2). UEZ nitrate gradients were ~0, constraining vertical nitrate fluxes to ≪ 0.01 μmol N m$^{-2}$ d$^{-1}$ (Supplemental Fig. 2). Although vertical diffusivities can vary significantly in space and time, the stratification strength and vertical diffusivities for our field study are consistent with independent estimates from the NEMURO-GOM biogeochemical model and with previously reported values[26].

As vertical mixing of nitrate and N$_2$-fixation individually provide enough N to support <1% of sinking PON flux in our Lagrangian experiments, locally sourced new N appears insufficient to balance the measured rates of PON export. Instead, a significant role of lateral transport of organic material is suggested by the rapid horizontal displacements in our drifter experiments (up to 60 km d$^{-1}$) and the strong coastal-offshore particulate gradients in the GoM (Supplemental Figs. 3–4). We tested this hypothesis with independent approaches of satellite remote-sensing products and a biogeochemical model tuned to the open-ocean GoM.

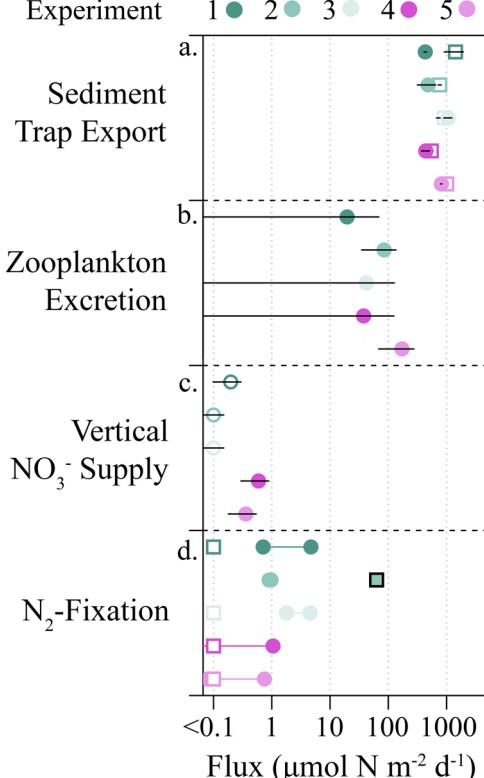

**Fig. 2 Summary of observed euphotic zone nitrogen fluxes from the oligotrophic Gulf of Mexico. a** Sediment trap fluxes include export at the base of the upper euphotic zone (UEZ) (open squares) and lower euphotic zone (LEZ; closed circles). **b** Zooplankton excretion below the euphotic zone. **c** Vertical mixing of nitrate into the LEZ. Mixing into UEZ not shown (~0). **d** N$_2$-fixation rates modeled from observed *Trichodesmium* biomass (UEZ; circles) and from δ$^{15}$N-based budgets for the entire euphotic zone (squares) as reported in Knapp et al[15]. Fluxes < 0.1 μmol N m$^{-2}$ d$^{-1}$ are shown on left as open symbols. Colors correspond to experiment as indicated with error bars indicating ±1 SD of measurement means (**a–c**) or reported ranges (**d**).

**Lateral supply of nitrogen**. MODIS Aqua[27] and Terra[28] remote-sensing products provide spatial POM data to infer lateral gradients in the GoM. Combined with a remote-sensing circulation product, OSCAR[29], we calculate lateral fluxes of particulate organic nitrogen (PON) in the UEZ. Independently, results from a 3D, nitrogen-currency biogeochemical model (NEMURO-GOM[30]) are used to resolve lower trophic level dynamics in the open-ocean GoM. Notably, NEMURO-GOM was developed to accurately simulate zooplankton dynamics (and hence was primarily compared to zooplankton observations). The MODIS POM and OSCAR circulation products were not used by NEMURO-GoM.

The interannual median UEZ lateral transport of PON (i.e., organisms and detritus) into the central GoM zone (Figs. 1a and 3) is estimated to be equivalent to a flux of 1150 μmol N m$^{-2}$ d$^{-1}$ (IQR: 610–1530 μmol N m$^{-2}$ d$^{-1}$) and 1165 μmol N m$^{-2}$ d$^{-1}$ (IQR: 700–1725 μmol N m$^{-2}$ d$^{-1}$) by remote-sensing and NEMURO-GOM, respectively. Both estimates (Fig. 3) indicate that lateral transport of PON alone is sufficient to support sediment trap-derived export out of the UEZ (median: 980 μmol N m$^{-2}$ d$^{-1}$; range: 587–1526 μmol N m$^{-2}$ d$^{-1}$) and the LEZ (median: 520 μmol N m$^{-2}$ d$^{-1}$; range: 462–1144 μmol N m$^{-2}$ d$^{-1}$). DON contributed an additional −8 – 46% to total UEZ lateral transport within NEMURO-GOM for any given day (Fig. 3), while lateral transport of DIN (nitrate+ammonium), generally found in diminishingly low concentrations in these waters (Fig. 1a; ref. [15]), was uniformly low in the UEZ and represented ~5% of net nitrogen transport regionally. Substantial agreement between these two distinct types of models is remarkable considering that even the physical circulation fields are unique to each model (global HYCOM for NEMURO-GOM; OSCAR circulation for satellite models). Taken together, these results clearly indicate that net lateral transport of organic matter is of sufficient magnitude to support open-ocean export in the GoM.

Circulation patterns in NEMURO-GOM suggest that the open-ocean GoM is predominately a downwelling region and thus an organic nitrogen sink. In the absence of significant vertical input of subsurface nitrate into the LEZ or recently fixed nitrogen into the UEZ, particulate N export can only be supported by either lateral transport or a non-steady state drawdown of bioavailable

nitrogen. Since the latter was not observed during any of our five, 4-day Lagrangian experiments (Supplemental Fig. 1c–e), lateral input is likely to be responsible for ~100 % of export out of the UEZ and >90% of export out of the LEZ (Fig. 4). While episodic transport events such as storms have been shown to be relevant in other regions[31,32], the impact of such transient processes on the N balance of open-ocean GoM is likely negligible in late spring and summer due to the very deep nitracline (80–125 m). Such second-order processes are likely more important near the shelves where nitracline and mixed layer depths are closer than in the experimental sites themselves. What is clear in the open-ocean GoM is the prevalence of mesoscale eddies formed through instabilities in the Loop Current[33] (Supplemental Figs. 3 and 4). Previous studies have found that, through both enhanced surface velocities and long persistence, such mesoscale features can transport large volumes of water onto and off of the shelf[34–36].

These lateral fluxes likely have implications for the entire GoM food web. Organismal carbon budgets suggest that carnivorous metazooplankton in the GoM may rely on subsidies of prey advected from the coast/shelf break region[37]. An individual-based model developed for Atlantic bluefin tuna larvae suggests that shelf break regions with strong offshore flow may be particularly important spawning locations that allow first-feeding larvae to find sufficient prey while transporting older larvae to low-predator regions[38]. Indeed, back-trajectories showed that locations with high larval abundance on the BLOOFINZ-GoM cruises were associated with water recently advected from more productive shelf break regions[39]. Thus, while much work is needed to elucidate relationships between specific taxa and nitrogen sources to the ecosystem, preliminary evidence shows that lateral transport is crucial to multiple trophic levels in the GoM.

**Global connections and implications.** Since the oligotrophic GoM has previously been compared to the mid-ocean gyres due to similar biogeochemical properties (e.g., DCM, oligotrophic, low biomass)[20,30,40], it may be reasonable to consider the applicability of our results to these oceanic regions. The mid-ocean gyres are both larger than the GoM and may have less average kinetic energy (leading to circulation time-scales of months-years, rather than weeks). Consequently, lateral PON fluxes are likely weaker in gyres but may be compensated, in part, by greater DON flux contributions[7]. Indeed DON inputs from lateral advection, entrainment, and mixing have been shown to supply gyre interiors[41,42], and based on model results may support 40–70% of export production in the North Atlantic gyre[43]. Furthermore, export efficiency (i.e., export/NPP) is lower in the subtropical gyres (~5%) than measured in our study (11–25%) indicating a reduced lateral input requirement to support gyre export. Taken together, these analyses suggest that lateral transport of organic matter could potentially be a dominant source of external nitrogen in the oligotrophic gyres, as it is in the GoM.

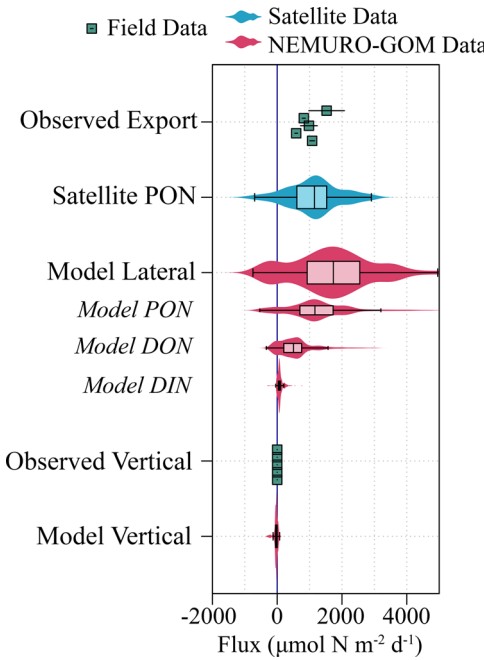

**Fig. 3 Comparison of lateral nitrogen supply to observed export production in the upper euphotic zone.** Satellite lateral particulate organic nitrogen (PON) flux was calculated from 8-day composite Moderate Resolution Imaging Spectroradiometer (MODIS) Terra/Aqua imagery during the month of May (2002–2019) with flow fields prescribed by OSCAR circulation. NEMURO-GOM fluxes were calculated separately for each nitrogen pool (italicized) and summed for total lateral fluxes. NEMURO-GOM vertical fluxes are integrated for the upper euphotic zone and include upwelling and turbulent mixing but not export by sinking particles. Positive flux values indicate net input into the integration volume. Flux values are normalized to lateral area. Boxes were determined from the 25th, 50th, and 75th percentile of the data, with whiskers indicating the limits of data extending 1.5× the interquartile range of the data.

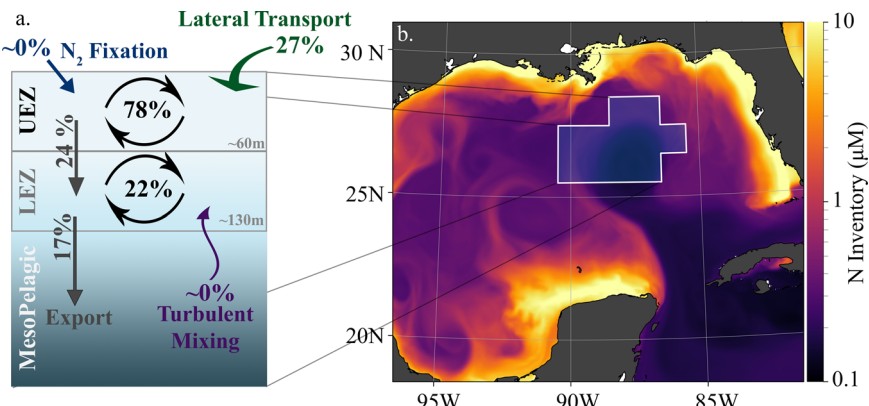

**Fig. 4 Schematic of transport mechanisms and fluxes of the northeast Gulf of Mexico. a** Arrows and values and fluxes observed or estimated from model relative to euphotic zone net primary productivity including turbulent mixing (purple), N₂-fixation (blue), lateral transport (green), gravitational sinking (gray) and primary production (black). Upper and lower euphotic zones are indicated. **b** Monthly surface snapshot of nitrogen inventories for May 1993 in NEMURO-GOM.

Our results highlight the importance of inter-disciplinary and integrative process studies for constraining biogeochemical fluxes. Among still unresolved issues, it is not clear whether the laterally transported organic matter was originally supported by nitrate from the shelf, upwelling at the shelf break, upwelling associated with mesoscale features, or from some combination of these. Furthermore, extrapolating our results to other time periods and conditions (e.g., winter) in the GoM should be done with caution and appropriate field measurements. Nevertheless, our results clearly highlight the importance of accounting for contributions of lateral transport to carbon and nitrogen budgets in oligotrophic ocean regions and the role that horizontal mesoscale currents have in supporting vulnerable open-ocean GoM ecosystems.

## Methods

**Sampling plan and environmental sampling.** Data were collected during two cruises in the Northern Gulf of Mexico conducted in May 2017 and April/May 2018[39]. Each cruise began with a bongo survey of water parcels that were identified by remote sensing as having favorable SST and vorticity[12] for larval Atlantic Bluefin Tuna (ABT). At selected locations, a sediment trap array was deployed and a Lagrangian experiment was initiated. Over the following 3–4 days, the water parcel was repeatedly sampled for biogeochemical properties and incubations were conducted to measure process rates. Throughout an experiment, profiles of temperature and salinity were taken 3–4 times per day through the upper 200 m. Additionally, PAR profiles were conducted at least once per Lagrangian experiment around local noon. Ambient nutrient concentrations were determined from samples collected daily from 6 depths throughout the euphotic zone (same depths as incubations were conducted). The concentrations of nitrate+nitrite were measured using a chemiluminescent method, while concentrations of $NH_4^+$ were measured using the fluorescent OPA method. See Knapp et al.[15] for details.

*Trichodesmium abundance and $N_2$ fixation.* Trichodesmium abundance and chlorophyll contents were assessed as reported in Selph et al.[20]. Briefly, 6.6 L samples from daily noon casts at 6 depths (2–50 m) were collected and filtered through inline 8-μm, 47-mm filters, preserved (2% paraformaldehyde), and frozen (−80 °C). Trichodesmium were counted with microscopy (10–30X magnification) under 440–460 nm excitation to detect orange-fluorescing trichomes. For chlorophyll a contents, separate samples (6.6 L) were collected onto 20 μm, 47-mm filters and frozen (−80 °C). Individual trichomes were picked from thawed filters, extracted (90% acetone), and chlorophyll fluorescence measured with 10-AU (Turner Designs) fluorometer[44]. Chlorophyll-specific $N_2$-fixation rates were calculated based on the photo-fixation model of Breitbarth et al.[21]) based on culture work conducted at temperatures (i.e., 26 °C) and photoperiods (i.e., 12 h) consistent with in situ conditions observed.

**Productivity and nutrient dynamics.** Rates of net primary production (NPP) and nitrate uptake were measured at 6 depths spanning the euphotic zone each day of a Lagrangian experiment. Incubation bottles (2.7 L) were filled by Niskin rosette and then $H^{13}CO_3^-$ (final concentration: 154 or 196 μmol $L^{-1}$ on NF1704 and NF1802, respectively) and/or $^{15}NO_3^-$ (final concentration: 10 or 8 nmol $L^{-1}$ on NF1704 and NF1802, respectively) was added. Bottles were then incubated in mesh bags attached to an in situ drift array[45] for 24 h—drift array was deployed and recovered pre-dawn, typically ~0500 local time. Upon recovery, incubation bottles were immediately filtered onto precombusted GF/F. Filtered samples were kept at −80 °C until analysis. In addition to in situ incubations, a series of deck-board experiments were conducted for 4–6 h to measure vertical patterns and diel variability in nitrate (same $NO_3^-$ spike as above) and ammonium uptake[17] (final concentration: 9.8 or 4.6 nmol $L^{-1}$ on NF1704 and NF1802, respectively). Samples were analyzed at the UC Davis stable isotope facility. Nutrient uptake rates and associated uncertainty were calculated as in Stukel[46].

*Thorpe scale analysis.* Profiles of vertical diffusivity ($K_z$) were calculated by Thorpe scale analysis[25] of CTD-derived density profiles. Thorpe scale analysis estimates turbulent mixing parameters (i.e., vertical eddy diffusivity, $K_z$) based on the number and size of observed density inversions. Raw Seabird data files were processed based on the recommendations of ref. [25]. Resulting profiles of $K_z$ were averaged across all casts during a Lagrangian experiment using the geometric mean ($n = 14$–26, with decreasing number of casts below the euphotic zone). Nitrate fluxes ($F_Z$) were then calculated using the observed vertical nutrient ($NO_3^-$) gradients and vertical diffusivity (Eq. 1).

$$F_z = -K_z \left( \frac{d[NO_3^-]}{dz} \right) \tag{1}$$

**Zooplankton abundance, biomass, and migrant excretion.** Zooplankton were collected daily from paired day–night tows with a 1-m diameter ring net (200-μm mesh) towed obliquely through the euphotic zone[16]. The tow contents were filtered through stacked Nitex screens of 5, 2, 1, 0.5, and 0.2-mm Nitex mesh to produce 5 size classes, and carbon, nitrogen, and dry mass for each size class were determined. Active nitrogen transport by daytime excretion of diel migrating mesozooplankton at mesopelagic depths was calculated from the equations of Ikeda[47], the biomass differences between paired day and night tows (size-fractioned migrant carbon), the average C values of individual animals in each migrant size fraction[48], and the mean temperature in the 300–500 m depth range (~10–12 °C).

**Export production by sediment trap.** Each sediment trap deployment consisted of three cross-members with 12 Particle Interceptor Tubes[49] per depth. Trap depths were determined from CTD fluorescence profiles with one cross member below the mixed layer and within the euphotic zone (60 m), one at the base of the euphotic zone (110–150 m), and one at 230 m. The sediment trap array included a 3-m x 1-m holey sock drogue at 15 m depth[50]. Sediment trap tubes were filled with a poisoned brine solution. After recovery, the overlying seawater was siphoned off to within 3 cm of the brine interface. The samples were then filtered through a 100-μm filter which was manually inspected under a stereo-microscope (25X) to remove zooplankton swimmers. After washing the >100 μm non-swimmer fraction back into the sample, whole tube contents were filtered onto a pre-combusted GF/F for C and N isotopic analyses.

## Analysis

*Nitrogen isotopic signatures of new and export production.* Mass balance requires that the nitrogen $\delta^{15}N$ isotopic signature of export equals the isotopic signature of the source nitrogen. $N_2$-fixation supplies nitrogen with a $\delta^{15}N$ of approximately −2–0‰[51,52] and vertical mixing of subsurface nitrate would maintain the subsurface isotopic signature: 2.0–3.8‰[15]. Exported nitrogen (sinking + active transport) was observed to have a $\delta^{15}N$ of ~2.9–5‰.

*Remote-sensing estimates of transport.* Eight-day composites of surface POC concentrations estimated from Moderate Resolution Imaging Spectroradiometer (MODIS) Aqua & MODIS Terra were retrieved from the National Aeronautical and Space Administration's (NASA) data repository[27,28] for each May from 2000 to 2019. We assumed that particles had Redfield stoichiometry (106:16, C:N, mol: mol). Remote sensing fields were binned to 8-km x 8-km resolution prior to analysis to reduce noise. Regional circulation was prescribed from OSCAR[29], a data-assimilative, remote-sensing data product with 1/3rd degree spatial and 5-day temporal resolution. OSCAR circulation fields were mapped onto the remote-sensing grid using bilinear spatial interpolation and linear temporal interpolation. Fluxes between each grid cell were then calculated for each remote-sensing field ($n = 80$). The total net flux into the control volume was divided by the area of the control volume for comparison to vertical N flux measurements. Results are shown for the control volume as defined in Fig. 1a.

*NEMURO-GOM estimates of transport.* The biogeochemical model NEMURO-GoM was used to provide a consistent three-dimensional perspective on nutrient uptake dynamics in the study region. The model consists of 29 z-layers with a vertical resolution of 10 m in the upper 150 m and ~4-km horizontal resolution. NEMURO-GOM is run offline within the MIT general circulation model and forced by daily-averaged flow fields obtained from a data-assimilative HYbrid Coordinate Ocean Model GoM simulation. The model has been extensively validated against nutrient, biomass, and rate measurements and fully resolves a simple nitrogen cycle with 11 state variables consisting of two phytoplankton and three zooplankton functional groups[30]. Vertical mixing in the model is parameterized based on a nonlocal K-profile parameterization (KPP) mixing scheme from a bulk Richardson number approach that quantifies the importance of stratification and destabilizing shear[53]. Daily model output for May of 1993 to 2012 were analyzed ($n = 571$) with flux integrations performed for the same control volumes as used for the remote sensing estimations (Fig. 1a). Mesoscale eddies were identified in daily-averaged model flow fields using the algorithm of Laxenaire et al.[54], which utilizes velocities along closed contours of SSH to define eddy boundaries. Lateral eddy flux (see Supplemental Discussion) was determined by calculating the net flux into the control column for all model grid cells that were identified as located within an eddy (i.e., enclosed SSH contour with the greatest velocity).

## Data availability

The data that support the findings of this work are available from the cited and publicly available repositories, including MODIS (NASA), OSCAR (ESR), and field data (BCO-DMO), or upon request to the author (NEMURO-GOM).

## Code availability

Codes used in the preparation of this manuscript are freely available under a Creative Commons license at the following url: https://github.com/tbrycekelly/Lateral-connectivity-in-the-gom, ref. [55], and from the author upon request.

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

## Acknowledgements
We thank the captain and crew of the NOAA Ship *Nancy Foster* for providing their assistance in conducting the fieldwork that led to this manuscript. We also thank our collaborators and colleagues at NOAA Southeast Fisheries/RSMAS/University of Miami for helping plan and carry out two successful field campaigns. We would like to thank Remi Laxenaire for his assistance in the identification of eddies. This study acknowledges BLOOFINZ Program support from National Oceanic and Atmospheric Administration awards NA15OAR4320071 (CIMEAS), NA16NMF4320058 (JIMAR), NA15OAR4320064 (CIMAS), and U.S. National Science Foundation awards OCE-1851347 and OCE-1851558. We also acknowledge support from the National High Magnetic Field Laboratory NSF/DMR-1644779 and the State of Florida.

## Author contributions
T.B.K. and M.R.S. planned and synthesized the data presented here. T.B.K., M.R.L., K.E.S., T.A.S., R.K.T. and M.R.S. collected the field data. A.N.K. provided nutrient data for the analysis. T.A.S. designed the hydrodynamic model and provided output data. T.B.K. wrote the manuscript. All authors took part in editing and submission of the final manuscript.

## Competing interests
The authors declare no competing interests.
