## [Peer Review File · Nature Communications]

REVIEWER COMMENTS

Reviewer #1 (Remarks to the Author):

The main claim of the paper is that lateral fluxes of nutrients dominate the N budget of the GoM. This is an important finding because a long standing paradigm has been to study nutrient dynamics using a 1-D (vertical) framework. Although there have been a few previous studies pointing to the importance of lateral transport, the present study stands out because of the careful analysis of multiple data streams taken during the same time period. As a result the findings are particularly clean and clear.

It was a pleasure to read your manuscript on the importance of lateral advection of N for balancing the GoM N budget. The manuscript was clearly written and convincing.

I have only two minor comments that might improve the manuscript.

1) I understand that the measurements and N budget are essential for constraining the base of the food web in the Gulf of Mexico. The authors point out in passing, and the acronym for the field campaign (BLOOFINZ-GoM) suggest, that the work is relevant to endangered Bluefin tuna but do not make the connection explicit. Is there a connection? My understanding is that it is quite difficult to link primary production to fish population. Anyway, if the authors could say a bit more about this link I'm sure many readers would appreciate it.

2) I would also have liked to see a few comments about the independence, partial independence or lack of independence between the NEMURO-GOM model output and the Satellite data. Figure 3 shows that both estimates agree very well. Is the agreement surprising or is it built in to the budget because the satellite data was used to constrain the model, or because similar data products were used to constrain OSCAR or NEMURO-GOM etc. The authors should consider giving the readers a bit more guidance on how to interpret the agreement. Is the point simply that consistent budgets can be produced using a detailed accounting of the fluxes or are the two budgets independent, in which case the agreement should give us more confidence in either budget separately. A few words explaining to what extent both estimates rely on common parameterizations and data sets used for parameter calibration in the circulation and the biogeochemical state would be welcome.

Reviewer #2 (Remarks to the Author):

Review of "Lateral Advection Support Nitrogen Export in the oligotrophic Open-Ocean Gulf of Mexico" by Kelly et al

The manuscript reports low N₂-fixation and low diffusive flux of nitrate into the euphotic zone, which cannot support relatively high PON export determined using sediment traps in the open-ocean Gulf of Mexico. The authors then reconciled this discrepancy by attributing the difference to laterally-sourced organic nitrogen. The lateral transport of organic matter into oligotrophic oceans are not a novel idea, there are field and model studies published in recently years. This manuscript is an addition to reaffirm the lateral transport idea in the Gulf of Mexico. Overall, the manuscript is well-organized and fits the scope of the journal. However, there are a few problems that need to be clarified before publication.

1) The authors determined N₂ fixation, and vertical diffusivity, but did not mention atmospheric deposition, and lateral transport of inorganic nitrogen, which could be potential N source in the Gulf of Mexico.

2) The gravitational export of organic nitrogen was determined at the depth of 60 m and at the

bottom of the euphotic zone. Have the authors thought about sinking speed of PON? Where was the PON originated? How far has the PON travelled before caught by the sediment traps? I noticed that the authors discussed the uncertainty of the PON origin at the end of the paper. They should at least calculate a rough range based on current speeds and PON sinking speeds (can refer previous papers for an estimate of sinking speed.)

3) The units in the paper are not consistent.

Reviewer #1 (Remarks to the Author):

The main claim of the paper is that lateral fluxes of nutrients dominate the N budget of the GoM. This is an important finding because a long standing paradigm has been to study nutrient dynamics using a 1-D (vertical) framework. Although there have been a few previous studies pointing to the importance of lateral transport, the present study stands out because of the careful analysis of multiple data streams taken during the same time period. As a result the findings are particularly clean and clear.

We thank the reviewer for her/his time spent reviewing and suggesting improvements for this manuscript.

It was a pleasure to read your manuscript on the importance of lateral advection of N for balancing the GoM N budget. The manuscript was clearly written and convincing.

I have only two minor comments that might improve the manuscript.

1) I understand that the measurements and N budget are essential for constraining the base of the food web in the Gulf of Mexico. The authors point out in passing, and the acronym for the field campaign (BLOOFINZ-GoM) suggest, that the work is relevant to endangered Bluefin tuna but do not make the connection explicit. Is there a connection? My understanding is that it is quite difficult to link primary production to fish population. Anyway, if the authors could say a bit more about this link I'm sure many readers would appreciate it.

Given this recommendation we now include a paragraph highlighting the connection between this study and the broader BLOOFINZ project (lines 155-164).

“These lateral fluxes likely have implications for the entire GoM food web. Organismal carbon budgets suggest that carnivorous metazooplankton in the GoM may rely on subsidies of prey advected from the coast/shelf break region. An individual-based model developed for Atlantic bluefin tuna larvae suggests that shelf break regions with strong offshore flow may be particularly important spawning locations that allow first-feeding larvae to find sufficient prey while transporting older larvae to low-predator regions. Indeed, back-trajectories showed that locations with high larval abundance on the BLOOFINZ-GoM cruises were associated with water recently advected from more productive shelf break regions. Thus, while much work is needed to elucidate relationships between specific taxa and nitrogen sources to the ecosystem, preliminary evidence shows that lateral transport is crucial to multiple trophic levels in the GoM.”

2) I would also have liked to see a few comments about the independence, partial independence or lack of independence between the NEMURO-GOM model output and the Satellite data. Figure 3 shows that both estimates agree very well. Is the agreement surprising or is it built in to the budget because the satellite data was used to constrain the model, or because similar data products were used to constrain OSCAR or NEMURO-GOM etc. The authors should consider giving the readers a bit more guidance on how to interpret the agreement. Is the point simply that consistent budgets can be produced using a detailed accounting of the fluxes or are the two budgets independent, in which case the agreement should give us more confidence in either budget separately. A few words explaining to what extent both estimates rely on common parameterizations and data sets used for parameter calibration in the circulation and the

biogeochemical state would be welcome.

We thank the reviewer for her/his suggestion and have added several lines to help place our model results in context: “Notably, NEMURO-GOM was developed to accurately simulate zooplankton dynamics (and hence was primarily compared to zooplankton observations). The MODIS POM and OSCAR circulation products were not used by NEMURO-GoM.” (lines 120-123). And later state “Substantial agreement between these two distinct types of models is remarkable considering that even physical circulation fields are unique to each model (global HYCOM for NEMURO-GOM; OSCAR for satellite models)” (lines 134-136).

Overall the agreement between models is truly remarkable given the lack of overlapping data between NEMURO-GOM and MODIS. NEMURO-GOM was explicitly tuned and validated against zooplankton abundance and grazing rates in order to be used as a lower trophic level model for larval fish studies and individual based modeling.

Reviewer #2 (Remarks to the Author):

Review of “Lateral Advection Support Nitrogen Export in the oligotrophic Open-Ocean Gulf of Mexico” by Kelly et al

The manuscript reports low N₂-fixation and low diffusive flux of nitrate into the euphotic zone, which cannot support relatively high PON export determined using sediment traps in the open-ocean Gulf of Mexico. The authors then reconciled this discrepancy by attributing the difference to laterally-sourced organic nitrogen. The lateral transport of organic matter into oligotrophic oceans are not a novel idea, there are field and model studies published in recently years. This manuscript is an addition to reaffirm the lateral transport idea in the Gulf of Mexico. Overall, the manuscript is well-organized and fits the scope of the journal. However, there are a few problems that need to be clarified before publication.

We thank the reviewer for her/his time spent reviewing and welcome the suggested changes/edits.

1) The authors determined N₂ fixation, and vertical diffusivity, but did not mention atmospheric deposition, and lateral transport of inorganic nitrogen, which could be potential N source in the Gulf of Mexico.

A note on atmospheric deposition of bioavailable nitrogen is now included in the supplemental discussion (line 69) on the source of nitrogen. Previous work (see Howe et al. 2020 Journal of Geophysical Research Oceans and references within) has found that atmospheric N deposition in the GoM is low relative to other sources including rivers, submarine groundwater discharge, N₂ fixation, and subsurface nitrate. Furthermore, low *Trichodesmium* biomass has been linked to low atmospheric iron supply (see Walsh and Steidinger, 2001; Journal of Geophysical Research) and low atmospheric deposition generally.

DIN was explicitly included in the NEMURO-GOM analysis (Figure 3) and we now include the statement “lateral transport of DIN (nitrate+ammonium) was uniformly low in the UEZ and represented ~5% of net nitrogen transport regionally” (lines 131-134).

Lateral transport of DIN was assumed to be negligible in the field data (Lagrangian sampling) as a result of uniformly low DIN concentrations in the mixed layer. DIN lateral transport could not be assessed through remote sensing, because it is not (currently) possible to estimate DIN through remote sensing.

2) The gravitational export of organic nitrogen was determined at the depth of 60 m and at the bottom of the euphotic zone. Have the authors thought about sinking speed of PON? Where was the PON originated? How far has the PON travelled before caught by the sediment traps? I noticed that the authors discussed the uncertainty of the PON origin at the end of the paper. They should at least calculate a rough range based on current speeds and PON sinking speeds (can refer previous papers for an estimate of sinking speed.)

Yes, we spent some time considering how the observations related back to particle composition and sinking rates out of the euphotic zone. Previous studies (e.g., McDonnell et al. 2010) have found average settling velocities of $\sim 100 \text{ m d}^{-1}$ for sinking particles, which would imply relatively rapid removal of actively-sinking particles from the euphotic zone (order of one day). However, we caution that such a calculation would assume that a single class of particles sinks from the upper euphotic zone through the lower euphotic zone and into the mesopelagic realm. This is not, however, what our data suggests. The nature of sinking particles (as assessed through C:N ratios, isotopic ratios, chlorophyll and phaeopigment contents, etc.) changed substantially with depth. This suggests that sinking particles were consumed and/or remineralized as they sank and new particles (of possibly very different types) were created. For additional discussion, see Stukel et al. (2021, doi: 10.1093/plankt/fbab001), which conducted detailed analyses of sediment trap results from our cruises.

Given these important points, we believe that it would be far too speculative for us to offer estimates of the horizontal distances traveled by sinking particles (i.e. the “statistical funnel” of Siegel et al.). By limiting our analyses here to only the introduction and export of nitrogen, we are able to use a simplified control volume approach to allow us to explicitly test our hypothesis. However, for a more in-depth analysis of Lagrangian transport in the GoM, we direct the reviewer to a preprint of Gerard et al. (attached, in review) wherein lateral transport rates are discussed.

3) The units in the paper are not consistent.

We agree with the reviewer that consistency among units are important for making clear comparisons between values. We have updated the N_2 -fixation units to be in $\mu\text{mol N m}^{-3} \text{ d}^{-1}$ to be more consistent with the flux units $\mu\text{mol N m}^{-2} \text{ d}^{-1}$ (lines 81, 93). We also changed from units of mmol to μmol on line 77.

REVIEWERS' COMMENTS

Reviewer #1 (Remarks to the Author):

I believe that the authors have adequately addressed my suggestions, and I have no further comments. I commend the authors for a thorough study and for submitting a well-written paper.

Reviewer #2 (Remarks to the Author):

I am happy with the revision. Publish it.